# Efficient Out-of-Distribution Detection in Digital Pathology Using Multi-Head Convolutional Neural Networks

**Jasper Linmans**[1]                                     JASPER.LINMANS@RADBOUDUMC.NL
[1] *Department of Pathology, Radboud University Medical Center, Nijmegen, the Netherlands*
**Jeroen van der Laak**[1]                               JEROEN.VANDERLAAK@RADBOUDUMC.NL
**Geert Litjens**[1]                                       GEERT.LITJENS@RADBOUDUMC.NL

## Abstract

Successful clinical implementation of deep learning in medical imaging depends, in part, on the reliability of the predictions. Specifically, the system should be accurate for classes seen during training while providing calibrated estimates of uncertainty for abnormalities and unseen classes. To efficiently estimate predictive uncertainty, we propose the use of multi-head CNNs (M-heads). We compare its performance to related and more prevalent approaches, such as deep ensembles, on the task of out-of-distribution (OOD) detection. To this end, we evaluate models trained to discriminate normal lymph node tissue from breast cancer metastases, on lymph nodes containing lymphoma. We show the ability to discriminate between in-distribution lymph node tissue and lymphoma by evaluating the AUROC based on the uncertainty signal. Here, the best performing multi-head CNN (91.7) outperforms both Monte Carlo dropout (88.3) and deep ensembles (86.8). Furthermore, we show that the meta-loss function of M-heads improves OOD detection in terms of AUROC.

**Keywords:** uncertainty estimation, digital pathology, multi-head ensembles.

## 1. Introduction

Motivated by the increased availability of data and computational power, numerous deep learning models are being developed for medical imaging. Due to the potential implications when applying such methods in a clinical setting, it is vital to report meaningful confidence values in addition to the predicted class label, allowing practitioners to asses the quality of the results being reported. This is especially important in digital pathology which is characterised by large amounts of possible anomalies. These anomalies range from insignificant scanning artefacts, which can be safely ignored, to clinically relevant abnormalities such as rare disease events, which should be flagged by low confidence predictions. However, conventional deep learning methods are unable to correctly associate anomalous data with meaningful low confidence values (Guo et al., 2017; Ovadia et al., 2019).

In efforts to correctly asses the confidence, i.e. the uncertainty of deep neural networks, various methods have been developed. A natural approach is based on post hoc calibration of softmax probabilities on a validation set through temperature scaling (Guo et al., 2017). Although this demonstrates well-calibrated predictions on a test set, the same is not guaranteed for conditions of distributional shift (Ovadia et al., 2019). Other methods, such as Monte Carlo dropout (MC-dropout) or deep ensembles, consider statistics from a predictive distribution, instead of a point prediction (Gal and Ghahramani, 2016;

Lakshminarayanan et al., 2017). Although effective at estimating uncertainty in controlled conditions of computer vision datasets (Ovadia et al., 2019), these methods suffer from computational complexity by requiring multiple training runs or forward passes at inference.

To enjoy high-quality uncertainty estimates while alleviating problems of computational complexity, we propose the use of multi-head CNNs (M-heads)[1] (Lee et al., 2015; Osband et al., 2016; Rupprecht et al., 2017). This method has similarities to deep ensembles (Lakshminarayanan et al., 2017), but it overcomes the burden of multiple training and inference runs due to extensive parameter sharing. Specifically, parameters in the early layers of a network are shared such that an ensemble is defined by a collection of randomly initialised final layers: the heads. A big advantage of such a low-cost ensemble is the ability to train all members simultaneously, allowing modifications to the loss function to promote diversity.

A rigorous setup for evaluating uncertainty estimation methods is to assess their performance on the task of out-of-distribution (OOD) detection. The idea here is to define two datasets: $\mathcal{D}_{in}$ containing samples similar to those seen during training and $\mathcal{D}_{out}$ containing OOD samples. The latter may contain unseen abnormalities from the same domain or a completely different domain. The task is to correctly discriminate between samples from $\mathcal{D}_{in}$ and $\mathcal{D}_{out}$ based on the uncertainty signal, i.e. without specific training. Due to the relevance of detecting OOD samples, some existing methods exclusively focus on anomaly detection (Zhai et al., 2016; Golan and El-Yaniv, 2018). Here, we are interested in methods which perform well at a target task and estimate uncertainty as an auxiliary objective.

In order to evaluate the performance of the multi-head model, we first train it to detect breast cancer metastases in lymph node resections. The quality of the uncertainty estimates is evaluated by the ability to discriminate between $\mathcal{D}_{in}$: an independent dataset containing lymph node resections and $\mathcal{D}_{out}$: containing resections with presence of lymphoma. When checking sentinel lymph nodes for breast cancer metastasis, incidental discovery of lymphoma is rare: it was found in only 1.6% of the cases (Fox et al., 2010). This makes it a valuable case study to evaluate predictive uncertainty: clinically relevant and large-scale. Our contributions are as follows: **1)** We apply popular uncertainty estimation approaches on a real-world, large-scale dataset and show the ability to detect anomalous data in digital pathology. **2)** We demonstrate improved performance of M-heads in comparison to more cumbersome, often used approaches and **3)** we show that a diversity-promoting loss function, unique to the multi-head CNN, is important for improved OOD detection.

## 2. Methods

Given a dataset $\mathcal{D}_{train} = \{(\mathbf{x}_n, y_n)|\mathbf{x}_n \in \mathcal{X}_{in}, y_n \in \mathcal{Y}_{in}\}_{n=1}^{N_{train}}$ where $\mathcal{X}_{in}$ and $\mathcal{Y}_{in}$ define the in-distribution input and target space, we train a multi-head model $p_\Theta(\mathbf{y}|\mathbf{x})$ to produce a set of hypotheses of size $M$, i.e. a function $\mathcal{X} \mapsto \mathcal{Y}^M$. The task is to train the model such that its mean prediction generalizes well to a held-out dataset $\mathcal{D}_{test} = \{(\mathbf{x}_n, y_n)|\mathbf{x}_n \in \mathcal{X}_{in}, y_n \in \mathcal{Y}_{in}\}_{n=1}^{N_{test}}$. We will refer to generalization on $\mathcal{D}_{test}$ as the target task of the model. Our goal here is not to obtain new state-of-the-art results on the target task, but to perform comparable with the current state-of-the-art while simultaneously providing high-quality uncertainty estimates to discriminate between $\mathcal{D}_{in}$ and $\mathcal{D}_{out}$.

---

1. Code for reproducibility is available at https://github.com/JasperLinmans/m-heads, with a mirror at https://github.com/DIAGNijmegen/m-heads.

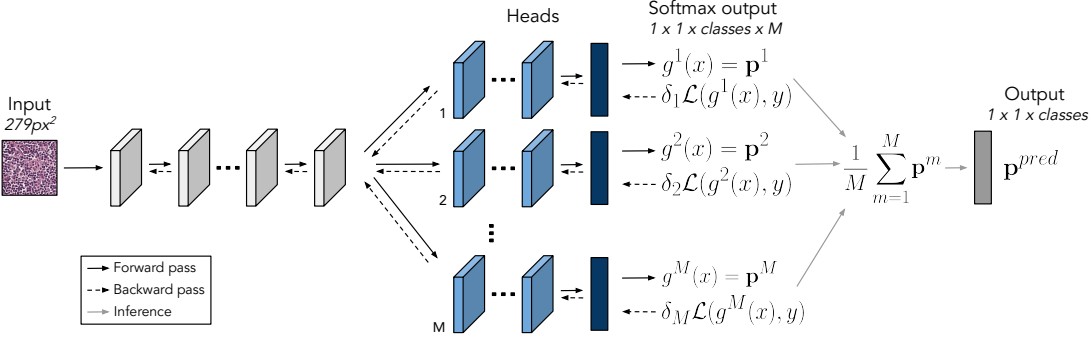

Figure 1: The multi-head convolutional neural network. Arrows denote the flow of operation through the network's shared part and the heads. During training, gradients are distributed following Equation (1). Predictions are averaged at inference.

## 2.1. Multi-Head Convolutional Neural Networks

To take advantage of the limited memory requirements of the multi-head model, we train all heads simultaneously using a meta-loss $\mathcal{M}$ (Rupprecht et al., 2017). See Figure 1 for an overview. The meta-loss function acts on top of any given standard loss, e.g. cross entropy, for a single data-point $(x, y)$:

$$\mathcal{M}(g(x),\ y) = \sum_{m=1}^{M} \delta_m \mathcal{L}(g^m(x), y) \tag{1}$$

where $g^m(x)$ is the softmax output of the $m$'th head, for all $M$ heads and such that

$$\delta_m = \begin{cases} 1 - \epsilon & \text{if } m = \arg\min_i \mathcal{L}(g^i(x), y). \\ \frac{\epsilon}{M-1} & \text{else.} \end{cases} \tag{2}$$

with $\epsilon$ the assignment relaxation constant. In other words, $\delta_m$ acts as a soft Kronecker delta such that a fraction $1 - \epsilon$ of the gradient signal flows through the head with the best hypothesis. The other heads receive the remaining signal. Reinforcing the performance of the winning head accordingly, will promote specialisation. Meanwhile, distributing the remaining loss will improve generalization of all heads to unseen data. We hypothesize that promoting diversity between heads using the meta-loss, will help to better cover lower density regions in the solution space which is beneficial for capturing ambiguities at inference. To prevent issues of mode-collapse, when training collapses to the predictions of a single head, we add stochasticity by randomly dropping out predictions with low probability (Rupprecht et al., 2017).

During inference, given an input $\mathbf{x}^*$, we use the M-head CNN to model the predictive distribution $p(y|\mathbf{x}^*; \Theta; \mathcal{D}_{train})$ where $\Theta$ are the parameters of the model. Let $\mathbf{p}^m$ denote the softmax probabilities produced by the $m$'th head, in a classification setting. We use the predictive mean across all heads to define the final prediction of the model: $\mathbf{p}^{pred} =$

$\frac{1}{M}\Sigma_{m=1}^{M}\mathbf{p}^m$. To asses predictive uncertainty we evaluate the entropy of the predictive mean across all classes:

$$\mathcal{H}[p(y|\mathbf{x}^*;\Theta;\mathcal{D}_{train})] = -\sum_{c=1}^{C} p_c^{pred} \log(p_c^{pred}) \tag{3}$$

## 2.2. Related Work

Due to a recent surge of interest, a variety of methods have been developed with the ability to provide estimates of uncertainty in addition to class predictions. Some of these approaches use statistics from the predictive distribution $p(y|x)$ to estimate uncertainty (Gal and Ghahramani, 2016; Lakshminarayanan et al., 2017). Others use an extra OOD detection component (Liang et al., 2017; Lee et al., 2018). We refer to (Ovadia et al., 2019) for a more systematic overview of existing approaches. Here, we focus on methods that use $p(y|x)$ directly because of their prevalence, scalability and similarities with M-heads.

A straightforward approach utilizes the confidence of a single model to signal uncertainty (Hendrycks and Gimpel, 2016). Here, given the predicted label $\hat{y}$, confidence is defined as the associated softmax probability $p(\hat{y}|\mathbf{x}^*;\boldsymbol{\theta})$. The majority of related work in medical imaging however, applies MC-dropout (Ghesu et al., 2019; Nair et al., 2020; Kwon et al., 2020). To model the predictive distribution, dropout is applied at test time and used in multiple stochastic forward passes through the network (Gal and Ghahramani, 2016). Previous work has shown that these uncertainty estimates can be used to detect anomalies in medical imaging (Leibig et al., 2017; Seebock et al., 2019). A different approach is the use of deep ensembles (Lakshminarayanan et al., 2017). Although this approach is similar to M-heads, due to the memory requirements of backpropagation, applying meta-loss objectives like Equation (1) to deep ensembles is practically infeasible. Instead, members are trained on different subsets of the training data, referred to as bagging, to promote diversity.

## 3. Experiments

The target task for this work is defined as the detection of breast cancer metastasis in whole-slide images (WSIs) of sentinel lymph node resections. To this end, each model is trained on data from the Camelyon16 challenge (Bejnordi et al., 2017). We will use the training and test set as defined by the challenge organizers. A proven method to train semantic segmentation models for WSIs is to adopt a patch-based classification approach. Here, models are trained on patches extracted from whole-slide images. Accordingly, we train each model on a set of 6M patches of size $279 \times 279$ from a $10\times$ resolution with a pixel spacing of $0.48\mu m$. Patches were extracted from the training-set in a ratio of $4:1$ normal to tumor, to reduce false positive detections, following previous work (Liu et al., 2017). To evaluate performance on the target task, we report the challenge metrics on the test-set: the FROC curve for tumor localization and the ROC curve for slide-level classification.

The performance for the task of OOD detection is evaluated using a dataset containing anomalies not seen during training. In particular, we consider 26 WSIs containing lymph node tissue diagnosed with diffuse large B-cell lymphoma. To compare the uncertainty estimates generated on these OOD samples with in-distribution samples independent from

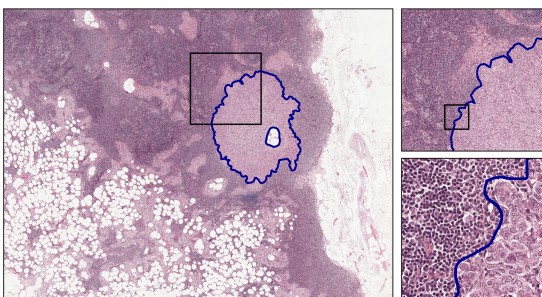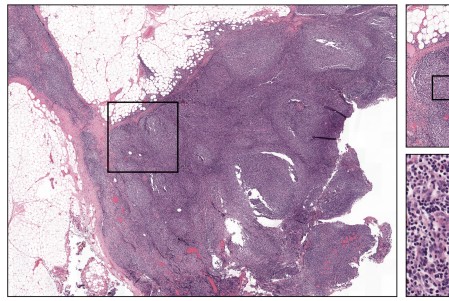

Figure 2: Images from $\mathcal{D}_{train}$ and $\mathcal{D}_{out}$ at different resolutions. **left:** Taken from $\mathcal{D}_{train}$ with it's ground-truth annotation. **right:** A WSI diagnosed with Lymphoma.

the training and test set of the target task, we use data from the Camelyon17 challenge. Specifically, we randomly selected 100 WSIs that were acquired by the same centers as Camelyon16, from the test set of Camelyon17.

At inference, entire WSIs are evaluated at pixel-level by leveraging fully convolutional networks[2]. However, only slide-level labels are available for both in-distribution and out-of-distribution data. Therefore, we perform spatial average pooling on the uncertainty heatmaps to produce a single uncertainty score per WSI. To do so, we apply a tissue-background segmentation algorithm to filter out the background (Bándi et al., 2019). Although there are more sophisticated methods for pixel-level to slide-level conversion, this is beyond the scope of this work. To compare the quality of the uncertainty estimates between models we evaluate AUROC for discriminating between $\mathcal{D}_{in}$ and $\mathcal{D}_{out}$. We also evaluate FPR at 95% TPR: the probability that an OOD sample is miss-classified as in-distribution, when the true positive rate (TPR) is as high as 95% (Liang et al., 2017).

**Training details:** In this work, we use a fully convolutional Densenet architecture (Huang et al., 2017), containing three dense blocks each with four valid padded convolutional layers. In total: 27 convolutional layers with 32 initial filters and a growth rate of 32. Each model is trained using the Adam optimizer (Kingma and Ba, 2015) for 50 epochs (defined as $2^{12}$ patches), with a batch size of 64 and an initial learning rate of 1$e$-4 (divided by 10 at 60% and 80% of training). The MC-dropout model additionally applies spatial dropout (Tompson et al., 2015) at the end of each bottleneck block (p=0.1). Based on results from (Ovadia et al., 2019), showing diminishing returns for larger sample sizes, we ran all MC-dropout evaluations using a sample size of 32. To study M-heads, we train models with 5 and 10 heads, where each head is defined by 4 convolutional layers and a classification layer ($\approx$ 15% of the convolutional layers of the DenseNet architecture). Here, $\epsilon = 0.05$ and a 1% probability for dropping the winning head were selected. To determine the importance of promoting diversity using the meta-loss, we also train models with a fixed Kronecker delta: $\delta_m = \frac{1}{M}$. For comparison we also include deep ensembles of size 5 and 10 (trained using a bagging approach). Finally, we include a baseline model: a single model where the uncertainty is defined by the entropy of it's softmax probabilities.

---

2. Here, we borrow terminology from Long et al. (2015)

## 4. Results

### 4.1. Performance on the target task

Table 1 shows the performance of each model on the target task: predicting cancer metastasis in whole-slide images of lymph node resections. Evaluations from a pathologist (Bejnordi et al., 2017) and results from Liu et al. (at 10× resolution) are included (Liu et al., 2017). Also, the results from the Camelyon16 winner are included, however, it uses 40× resolution and is thus not directly comparable (Wang et al., 2016).

Table 1: Results on the target task. Confidence bounds are obtained by 2000-fold boot-strapping. M-heads⁻ refers to the models trained with a fixed Kronecker delta.

| Method | FROC | AUROC |
|---|---|---|
| Baseline | 77.0 (67.6, 87.0) | 97.2 (93.9, 99.8) |
| MC-Dropout | 75.2 (64.5, 85.7) | 95.7 (90.5, 99.9) |
| Deep Ensembles (5) | 77.9 (67.1, 88.1) | 97.0 (93.0, 99.8) |
| Deep Ensembles (10) | 77.4 (66.4, 87.8) | 97.0 (93.0, 99.8) |
| M-heads⁻ (5) | 77.6 (66.6, 88.1) | 96.6 (92.2, 99.7) |
| M-heads (5) | 77.0 (66.2, 87.0) | 97.9 (95.0, 99.9) |
| M-heads⁻ (10) | 75.4 (64.0, 86.6) | 97.2 (93.5, 99.8) |
| M-heads (10) | 78.2 (67.5, 88.1) | 98.0 (95.3, 99.7) |
| Liu et al. | 79.3 (74.2, 84.1) | 96.5 (91.9, 99.7) |
| Camelyon16 winner | 80.7 | 99.4 |
| Pathologist | 73.3 | 96.6 |

These results show that all models are competitive with the current state-of-the-art. Furthermore, we observe that each model outperforms the single-network baseline, except MC-dropout and M-heads⁻ (10), which have a somewhat lower FROC score.

### 4.2. Performance on OOD Detection

We present the results of our main contribution, on the task of OOD detection in Figure 3. In terms of AUROC and FPR at 95% TPR, all uncertainty-based methods show consistent improvements over the baseline model, although MC-dropout has a slightly higher FPR. The M-head model with ten heads outperforms other models, followed by the M-head model with five heads. Training without the meta-loss (M-Heads⁻) decreases the performance of the multi-head model, but it still outperforms other approaches in terms of FPR and AUROC, except for MC-dropout on AUROC. Similar results are found when evaluating model disagreement instead of entropy, see Appendix A. We have included a visualisation of the predictions and uncertainty heatmaps produced by the M-head model for both in-distribution and OOD WSIs in Figure 4 (see Appendix B for a more extensive overview of randomly selected WSIs).

The entropy values used to calculate the results are visualised in Figure 5 (top row) and Figure 5 (c) for each individual model through their cumulative distribution (CDF). Here,

| Method | FPR at 95% TPR ↓ | AUROC |
|---|---|---|
| Baseline | 45.2 (25.1, 65.4) | 84.2 (77.5, 91.3) |
| MC-Dropout | 48.3 (26.9, 68.2) | 88.3 (81.5, 94.1) |
| Ensemble (5) | 42.4 (24.2, 61.3) | 86.8 (80.2, 92.7) |
| Ensemble (10) | 43.4 (24.0, 62.5) | 86.8 (79.9, 92.9) |
| M-heads⁻ (5) | 41.5 (25.0, 57.9) | 87.4 (81.1, 93.0) |
| M-heads (5) | 40.5 (22.7, 58.3) | 88.7 (82.7, 93.9) |
| M-heads⁻ (10) | 41.8 (25.0, 59.1) | 88.4 (82.0, 93.8) |
| M-heads (10) | 28.9 (12.0, 46.2) | 91.7 (86.3, 96.5) |

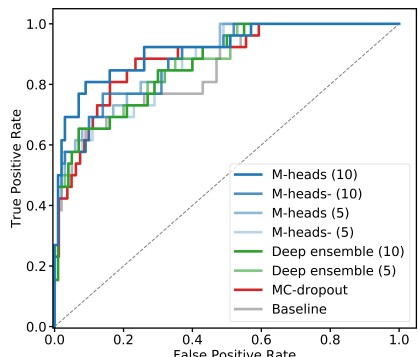

Figure 3: Performance on the task of out-of-distribution detection. Confidence bounds are obtained by 2000-fold bootstrapping. **right:** the corresponding ROC curves.

we see a slightly bigger discrepancy between in-distribution and OOD entropy values for the M-head models, which resulted in the improved AUROC values (Figure 3).

To get more insight into the differences in performance, we evaluate accuracy versus confidence plots (Lakshminarayanan et al., 2017) in Figure 5 (a, b), that were calculated using 200k patches extracted from the test set. Here, confidence is defined by the predicted probability associated with the predicted label. Accuracy values are calculated by filtering out predictions with confidence values lower then a particular confidence threshold $\tau \in [0, 1]$.

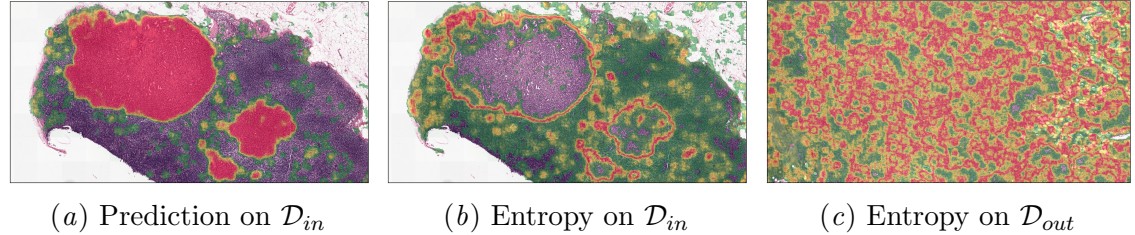

$(a)$ Prediction on $\mathcal{D}_{in}$      $(b)$ Entropy on $\mathcal{D}_{in}$      $(c)$ Entropy on $\mathcal{D}_{out}$

Figure 4: Output from M-heads (10). Predictions and entropy values from high to low: red, green and transparent. (a) The prediction on macro metastatic tissue from Camelyon17 with (b) the corresponding uncertainty. (c) A same-size entropy heatmap predicted on lymphoma tissue from $\mathcal{D}_{out}$.

## 5. Discussion

Our results show that all popular uncertainty-based methods, trained to detect breast cancer metastasis, are able to identify OOD whole-slide images diagnosed with lymphoma. Surprisingly, although in line with previous work (Hendrycks and Gimpel, 2016), the baseline defined by a single model also achieved reasonable OOD detection performance. Still, the uncertainty-based methods consistently improved on these results, with one exception

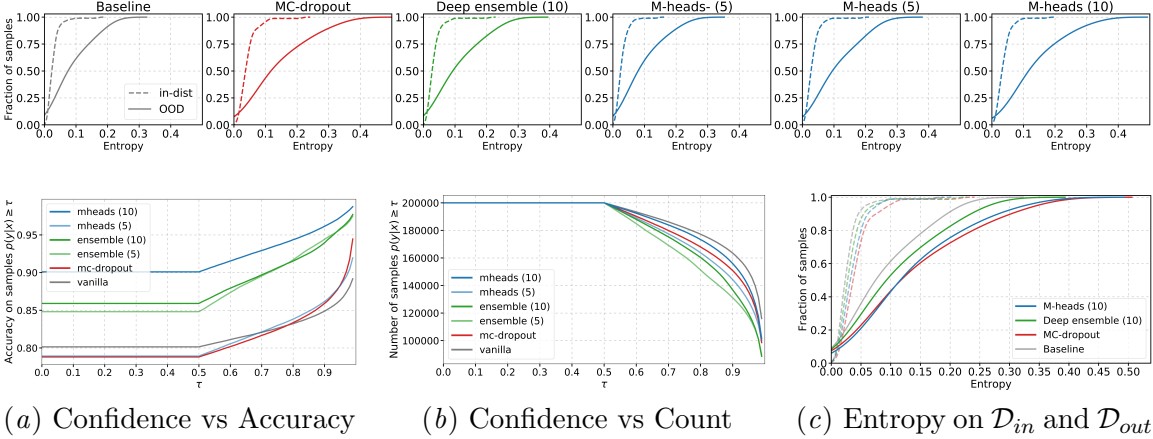

(a) Confidence vs Accuracy     (b) Confidence vs Count     (c) Entropy on $\mathcal{D}_{in}$ and $\mathcal{D}_{out}$

Figure 5: **top row:** CDFs of the predictive entropy for each model on both $\mathcal{D}_{in}$ and $\mathcal{D}_{out}$.
**bottom row:** (a, b) Confidence score vs accuracy and count respectively, evaluated for 200k patches from $\mathcal{D}_{test}$. (c) Summary of predictive entropy CDFs.

for the FPR of the MC-dropout model. We observe improved results for the multi-head approach compared to the other methods when trained using the meta-loss function, Equation (1). These results demonstrate the importance of the meta-loss function, which is not applicable to regular ensembles without significantly changing the network or the training procedure to deal with memory issues. We furthermore observe slightly better results on the target task for M-heads, in line with prior work (Lee et al., 2015). We hypothesize that the improved performance of M-heads is partially due to an increased diversity of it's predictions, which we demonstrate in Appendix A. Although, future work with a more extensive analysis on the diversity among the heads is required to confirm this.

Figure 5 shows that uncertainty based approaches are less confident in general and more accurate at higher confidence values compared to the baseline model, which suggests an improved calibration. This could partially explain the improved performance in OOD detection compared to the baseline model. For MC-dropout specifically, we see a lower target task performance compared to the other methods, indicating that applying dropout throughout the network could hurt performance. However, we see a comparable OOD detection performance with the more efficient multi-head approach based on five heads, which could be attributed to the diversity within the predictions (Appendix A).

In future projects, we would like to expand on this work by confirming the results on different tasks in digital pathology. Also, we would like to improve our pixel-level to slide-level conversion method, since the current approach gives sub-optimal slide-level scores due to uninformative estimates on fat tissue (examples in Appendix). Future work should also include a comparison with deep ensembles trained without bagging, which could potentially influence their OOD-detection performance. Furthermore, we hypothesize that disentangling our predictive uncertainty into epistemic and aleatoric uncertainty could help to increase the information provided by the uncertainty signal (Kendall and Gal, 2017). This could improve down-the-line tasks such as out-of-distribution detection.

## Acknowledgments

We would like to thank Jim Winkens for laying the foundation for this entire work during his Master's thesis.

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

## Appendix A. Disagreement in Predictions

In this work, we show improved out-of-distribution detection results for the M-heads approach compared to the other baselines. We hypothesize that this difference in performance is partially explained by the increased diversity between the individual predictions: between the heads in the case of M-heads. One way to analyze the diversity between individual predictions is to evaluate their disagreement. Therefore, to give additional insight into the differences in OOD detection performance, we will evaluate model disagreement as defined by (Lakshminarayanan et al., 2017):

$$\mathcal{D}([p_m(y|\mathbf{x})]_{m=1}^M) = \sum_{m=1}^M JSD(p_m(y|\mathbf{x})||p_e(y|\mathbf{x})) \tag{4}$$

where JSD denotes the Jensen-Shannon divergence and $p_e(y|\mathbf{x})$ the ensembled prediction: $p_e(y|\mathbf{x}) = \frac{1}{M}\Sigma_{m=1}^M p_m(y|\mathbf{x})$. Since the vanilla baseline is based on just a single prediction, it is excluded from this analysis.

Similar to the entropy of the predictive distribution, disagreement provides another useful quantitative way to evaluate predictive uncertainty. Therefore, we repeat the experiment as described in the method section of this work, but this time based on model disagreement instead of entropy, see Table 2. As shown by the results, the M-head models show a bigger discrepancy between in-distribution and OOD samples compared to the other methods, when trained using the meta-loss function. The increased diversity within the predictions on OOD samples could partially explain the improved OOD detection based on the predictive entropy (Figure 3). As expected, disagreement between the heads is less informative when M-heads is not trained with the meta-loss function. Interestingly, the disagreement within

the predictions of MC-dropout is more descriptive for OOD detection compared to the deep ensembles, which could explain the differences in performance as shown in Figure 3.

Table 2: Performance on the task of out-of-distribution detection based on model disagreement. Confidence bounds are obtained by 2000-fold bootstrapping.

| Method | FPR at 95% TPR ↓ | AUROC |
| --- | --- | --- |
| MC-Dropout | 44.6 (25.0, 64.3) | 86.0 (78.0, 93.0) |
| Ensemble (5) | 42.5 (24.0, 62.0) | 85.1 (77.5, 91.9) |
| Ensemble (10) | 44.8 (25.0, 64.0) | 83.4 (75.3, 90.7) |
| M-heads⁻ (5) | 50.9 (30.0, 70.8) | 81.5 (72.2, 90.1) |
| M-heads (5) | 42.4 (23.1, 63.0) | 87.6 (81.3, 93.2) |
| M-heads⁻ (10) | 43.5 (26.1, 60.9) | 81.3 (72.0, 89.7) |
| M-heads (10) | 38.9 (16.7, 61.5) | 87.3 (77.4, 95.7) |

## Appendix B. Uncertainty Heatmaps

Here we provide a random selection of whole-slide-images and the corresponding uncertainty heatmaps predicted by the M-head (10) model. For each figure: the left panel displays the input given to the network. Next to it, the uncertainty heatmap is shown, together with the slide-level uncertainty score (calculated by average pooling across the tissue). Here red values indicate high uncertainty whereas green and transparent indicates low values of uncertainty. The first four rows represent randomly selected out-of-distribution images (Figure 6). The final four rows show randomly selected in-distribution images (Figure 7).

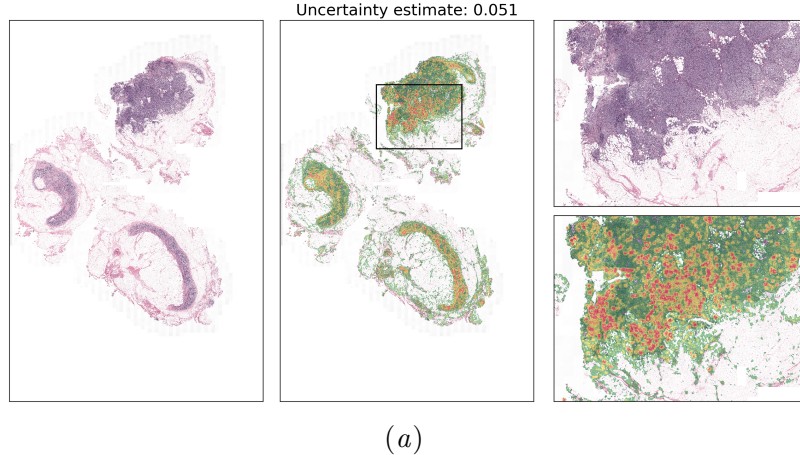

(a)

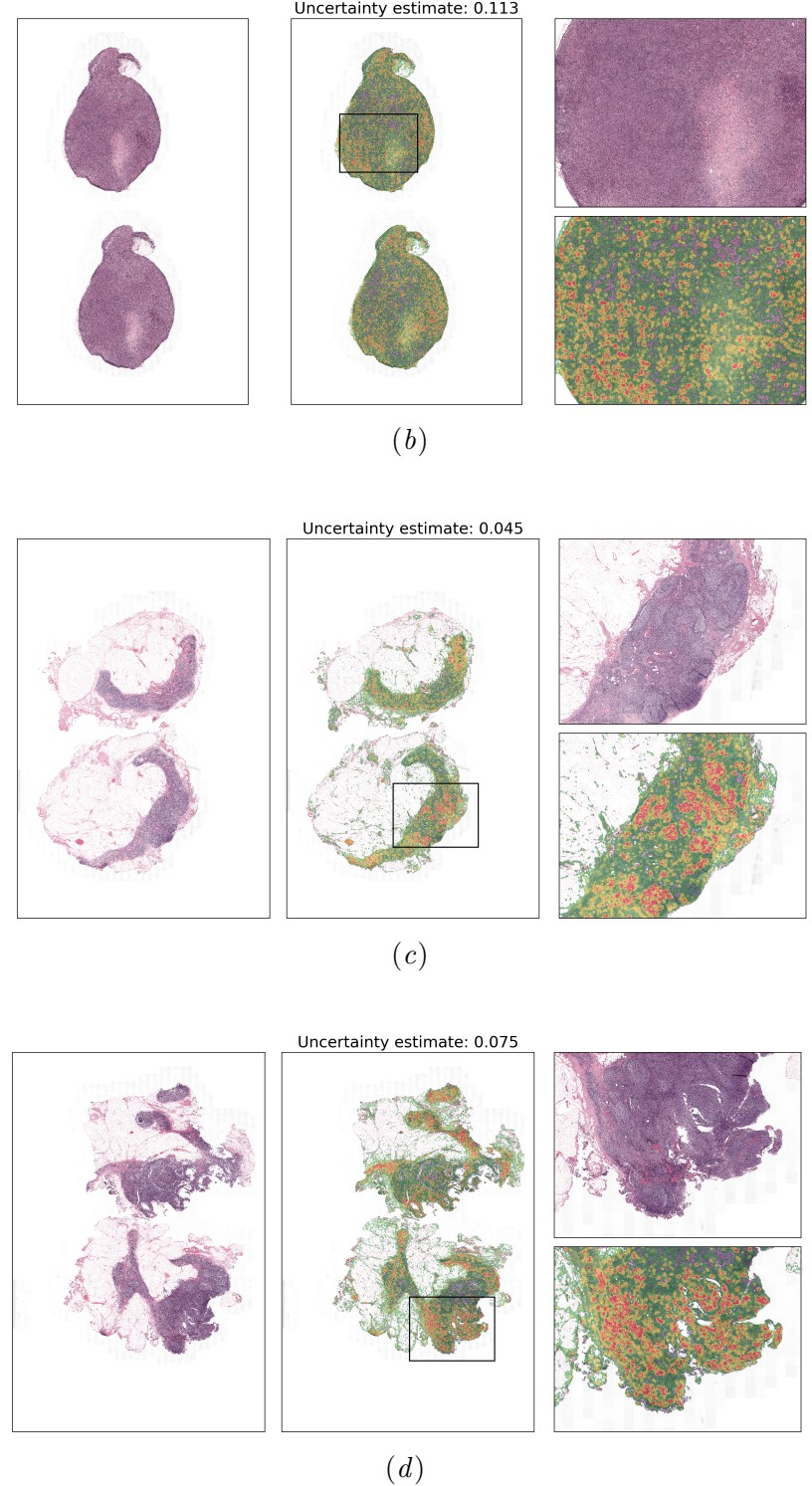

Figure 6: Four out-of-distribution WSIs and their corresponding uncertainty heatmaps, as produced by the M-Heads (10) model.

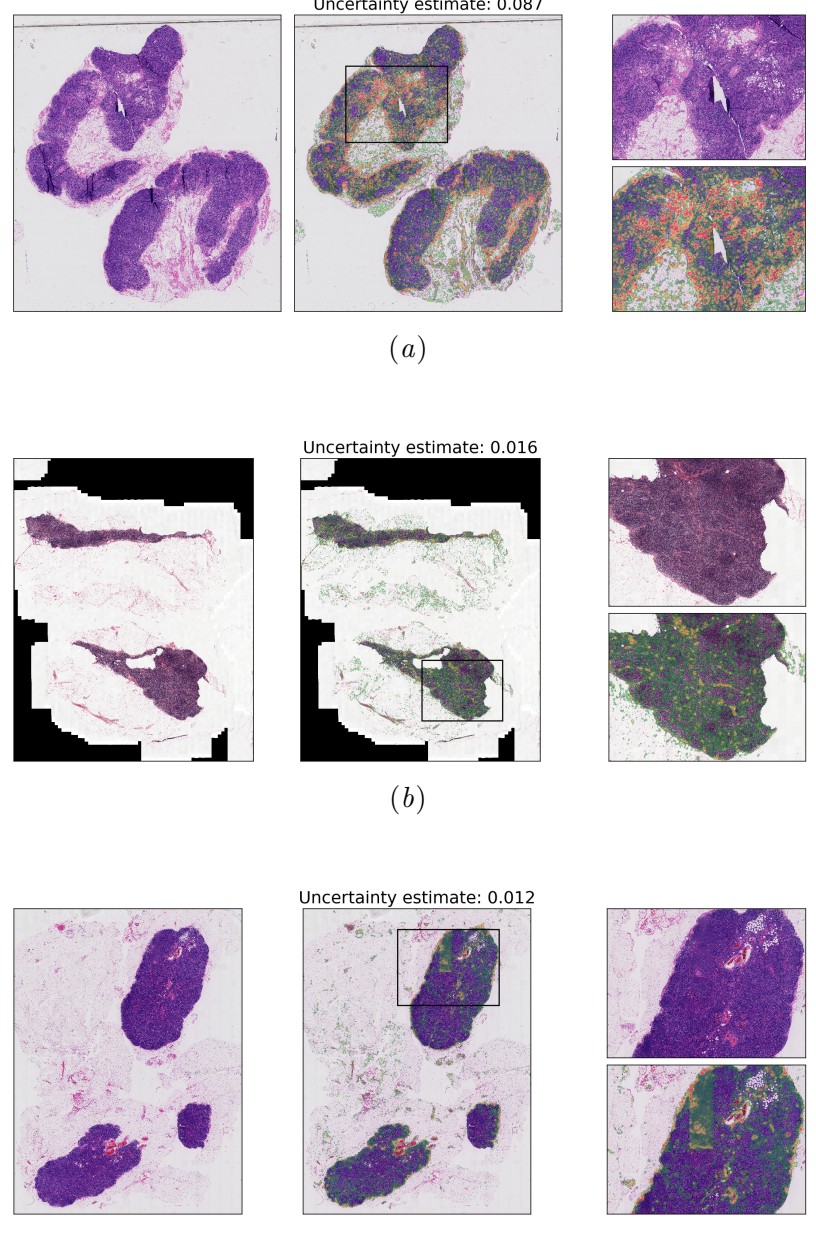

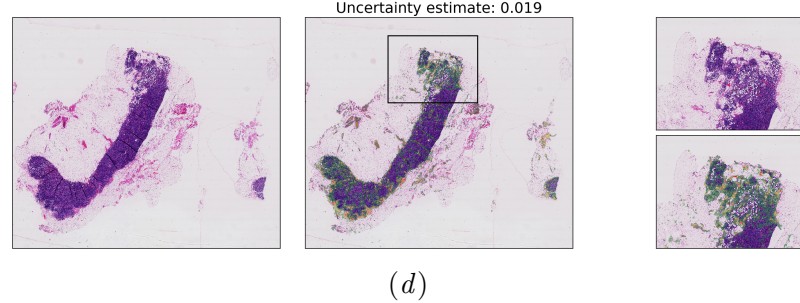

(*d*)

Figure 7: Four in-distribution WSIs and their corresponding uncertainty heatmaps, as produced by the M-Heads (10) model. The first row is an example of an in-distribution WSI with relatively high levels of uncertainty.

