# OpenReview forum: "Efficient Out-of-Distribution Detection in Digital Pathology Using Multi-Head Convolutional Neural Networks"
_MIDL.io/2020/Conference — MIDL 2020_

### Official Review · AnonReviewer1 · 2020-03-09
**Multi-head neural network for detecting outliers**

**Rating:** 2
**Confidence:** 4

**Summary:**

The paper introduces the use of multi-head neural networks for uncertainty quantification on digital pathology. The paper uses a meta-loss that shall induce diversity in the different heads. The proposed method achieves competitive performance and is capable of detecting outliers on the digital pathology task.

**Strengths:**

The paper applies a very resource efficient method for uncertainty prediction to digital pathology. It is simple to implement and shows competitive results on the target task. The method provides improved uncertainty estimation and out-of-distribution detection than a baseline model.

**Weaknesses:**

Even though the paper has an extensive validation, I am very cautious of the results from the ensemble baselines. The paper uses subsampling of the original dataset to encourage diversity of the neural networks. However, Lakshminarayan, et al. already showed that ensembles without subsampling but different initialisations perform well. This and the presented behaviour that an ensemble of 10 performs worse than an ensemble of 5 makes me question the quality of the baselines. The authors mention simple data subsampling according to Karimi, et al., however, Karimi, et al. seem to perform a data sampling strategy aimed at improving performance on difficult examples, which is very different from simple subsampling.
Further, the authors hypothesise that the improved performance of the multi-head approach is caused by increased diversity of the predictions which is induced by the meta-loss. I would improve the quality of the paper if the authors would report metrics like 'disagreement' of the predictions as proposed in Lakshminarayan, et al. This could also remove uncertainties of why the M-heads- baseline with 5 heads performs on par with M-heads with meta-loss. Further, it is mentioned that the meta-loss is practically infeasible to be used for ensembles. However, it should be possible to decrease the batch size and use train multiple models in parallel with the mentioned meta-loss. Both those comparisons would greatly improve the insights into the proposed and alternative methods. I would expect ensembles with that meta-loss to perform better, due to better coverage of the parameter space. However, the multi-head approach has significant resource benefits.

**Detailed Comments:**

- The authors mention (page 6) that the model displays aleatoric uncertainty in Fig 4. However, the model is not capable of disentangling aleatoric (data-specific) and epistemic (model-specific) uncertainty. M-heads is rather modelling the uncertainty in parameter space.
- Fig 1. and the equation on the bottom of page 3 seem like they are missing a 1/M as the prediction is the mean prediction.
- It might be worth-wile adding a reference to Osband, et al. Deep exploration via bootstrapped DQN, 2016 which is very similar to M-heads.

**Justification Of Rating:**

I believe that the paper proposes a valid and interesting method for uncertainty estimation and out-of-distribution detection on digital pathology. However, the paper does not introduce methodological novelty and misses important analysis in understanding the behaviour of the method and relevant baselines.

**Paper Type:**

validation/application paper

**Special Issue:**

no

---

> ### Author Response · Authors · 2020-03-27
> **Comments for AnonReviewer1. Included additional analysis on baseline (1/2).**
>
> We would like to thank the reviewer for his or her time during the reviewing process. We are happy that the reviewer highlights the efficiency of our approach and its improved out-of-distribution detection performance compared to other approaches. However, a few concerns are raised which we would like to address below. We are happy to discuss any remaining remarks with the reviewer.
>
> The reviewer is cautious of the results from the ensemble baseline because it has been trained using bagging methods. Lakshminarayanan mentions they achieved an increased performance using random initialisation instead of bagging. However, they also admit that bagging is a good mechanism to introduce diversity. We aim to compare the multi-head model, with its diversity promoting loss with the most competitive baselines on the task of out-of-distribution detection. We argue that it is fairer to compare our approach with a deep ensemble trained using bagging to minimize issues of mean-seeking behaviour, which would harm its ability to detect OOD samples. We would like to further emphasise that our goal is to compare these methods for the task of OOD detection and not on the target task which the reviewer is referring to. The target task is only to ensure each approach is properly optimized and the differences in performance here are negligible.
>
> We agree with the reviewer that adding deep ensembles trained using just random initialisations would be a valuable addition to this work. However, this is unfeasible given the limited time and computational resources for the rebuttal, even though we would be happy to add this additional comparison. Therefore we would like to address this issue threefold:
> 1. This work is based on a significant amount of data: 6 million patches. As mentioned by Lakshminarayanan, sampling with replacement (bagging) would keep around 63% unique patches in each subsample. Thus, every model in the ensemble is trained on almost 4 million unique patches. We would expect that this is more than enough to still sufficiently train each member of the ensemble to near SOTA performance. This is different from work by Lakshminarayanan, which is based on smaller datasets.
> 2. To convince the reviewer on this, we have performed inference using individual members of the deep ensemble on the target task. When comparing these results we see no difference with the vanilla baseline (which has been trained on all 6M patches).
>
> Target task performance (breast cancer metastasis detection):
> Model,                                          Bagging,    FROC,                            AUC
> Baseline model                           no               77.0 (67.6, 87.0)           97.2 (93.9, 99.8)
> Random ensemble member    yes              77.2 (66.8, 87.8)           97.2 (93.4, 99.8)
> Random ensemble member    yes              77.0 (66.5, 87.9)           97.5 (93.9, 99.7)
> Random ensemble member    yes              77.1 (66.0, 87.4)           96.6 (92.5, 99.7)
> Random ensemble member    yes              77.7 (67.7, 88.1)           96.3 (92.1, 99.5)
> Random ensemble member    yes              77.8 (67.1, 88.7)           97.6 (94.2, 99.9)
>
> 3. We have updated the discussion section of our submission to include a reference to Lee et al (2015). The authors here have produced similar results to our work on the target task, using classical computer vision datasets. Lee et al. show that multi-head-like models sometimes outcompete deep ensembles on classification tasks. This is true even when the deep ensemble is trained using random initialisations and without bagging.
>
> The reviewer mentions that the ensemble with 10 members is showing slightly lower performance in terms of FROC on the target task (although this is not reflected by the AUC) compared to the smaller ensemble. We think this might be because a smaller ensemble is already sufficient for optimal performance on the target task; increasing the ensemble size will not further improve its performance. These findings are in line with recent work by Ovadia et al. (2019) which found diminished returns when increasing the size of the ensemble above five.
>
> The review mentions that we incorrectly cite Karimi et al. when discussing data sampling strategies in our related work section. We agree that this is misjudged by us and we have dropped this work from our citation list accordingly.

---

> > ### Author Response · Authors · 2020-03-27
> > **Comments for AnonReviewer1. Included additional analysis on baseline (2/2).**
> >
> > Next the reviewer suggests reporting model ‘disagreement’ to give additional insights into the differences in performance. We agree that this can be used to further compare models on the task of OOD detection. We will include these results at a later time, well before the end of the discussion phase, in an additional comment below.
> >
> > The reviewer secondly suggests training an ensemble using the meta-loss function, which is only possible by reducing the batch size accordingly. Splitting our batch size of 64 over 5 or 10 ensemble members however, would lead to batch sizes of approximately 12 and 6 respectively. This would harm performance on both the target task and out-of-distribution detection due to training instabilities (especially due to the batch norm layers that are present in the network). Again, we would still be happy to include this comparison if time and computational resources would permit which is unfortunately not the case within the rebuttal period.
> >
> > Finally, the reviewer gives three additional detailed comments which all resulted in improvements to our submission. First, the author correctly addresses the fact that the multi-head model is currently not capable of disentangling aleatoric and epistemic uncertainty. Therefore, we have dropped this detail from our discussion. We have also included the missing 1/M term in both Fig. 1 and the corresponding equation. Thirdly, we have included the reference to Osband et al. as they provide a similar approach and use it for uncertainty estimation, in a setting of reinforcement learning.
> >
> > We thank the reviewer again for his or her comments and we are happy that they led to improvements to our submission. We hope to discuss any remaining remarks with the reviewer during the discussion phase.

---

> > > ### Comment · AnonReviewer1 · 2020-04-01
> > > **More clarification on behaviour of M-heads**
> > >
> > > Thanks for the clarifications and further insights. The authors addressed most of my concerns. However, I would like to better understand the behaviour of M-Heads in comparison to standard ensembles.
> > >
> > > From experience and literature (e.g. Ovadia et. al.) it seems that correctly trained deep ensembles are a very strong baseline if trained correctly. Therefore I am very surprised about the improved performance of M-Heads over deep ensembles and would like to better understand this behaviour. Thanks for the clarification of the single model performances. It would be very interesting to include model disagreements and compare those performances to 'single head' performances. Given your argumentation, the single head performance should be comparable to those baselines but exhibit a higher diversity / disagreement if I am not mistaken?
> > >
> > > I am aware that it is infeasible for to retrain new models within the discussion period. Nevertheless, I believe that the paper could benefit from additional insights disentangling the impact of M-Heads and the diversity loss. This includes both training a deep ensemble using the meta-loss as well as training M-Heads (10) without the meta-loss. Currently the performance of M-Heads-(5) and M-Heads(5) seems very similar and it is not clear to me whether that would hold for M-Heads(10). Further, it is argued that the increase from 5 to 10 ensemble components would produce diminishing returns. Should this not also be the case for M-Heads? As for training ensembles with small batch sizes - it is possible to use gradient accumulation to have larger virtual batch sizes to smoothen the training. This might incur issues with batch normalisation that could be resolved by using another form of normalisation (e.g. instance norm).
> > >
> > > Additional minor comments:
> > > - The reference to Ovadia should be updated to reflect the publication at NeurIPS 2019.
> > > - If time permits, I encourage the authors to experiment with changing the histograms in Fig 5. from pdf plots to cdf (cumulative density) plots. Bayesian deep learning literature has recently more often used CDF rather than PDF plots for the outlier detection as it shows the separation more clearly.

---

> > > > ### Author Response · Authors · 2020-04-04
> > > > **Included additional analysis on the behaviour of M-heads (1/2)**
> > > >
> > > > We would like to thank the reviewer again for his or her time and we are happy that we have addressed most of the concerns during the rebuttal phase. We would like to address the remaining remarks by including additional results below.
> > > >
> > > > The reviewer is surprised by the increased performance of M-heads compared to deep ensembles. We would like to stress again that the difference in performance on the target task, which is mainly used to verify that the models are properly optimized, are small. However, the fact that the multi-head model shows slightly better results here is in line with previous work on a similar model. Work by Lee et al. (2015) “Why M Heads are Better than One: Training a Diverse Ensemble of Deep Networks“ showed similar results using models they refer to as TreeNets: deep ensembles with shared parameters in the early layers. To quote the author: “We demonstrate that TreeNets can improve ensemble performance and that diverse ensembles can be trained end-to-end under a unified loss.” and: “ We see that shared parameter networks not only retain the performance of full ensembles, but can outperform them”. To partially address the concerns of the reviewer we will include this reference in our discussion section, as well as in the method section.
> > > >
> > > > However, to further validate our results and to give additional insights in the differences into performance, we will present three additional results which could be included in a camera-ready version.
> > > >
> > > > 1. The reviewer asks to analyse the performance of individual heads. We have analysed the results of a few random heads and have included the results below:
> > > >
> > > > Target task performance (breast cancer metastasis detection):
> > > > Model,                      FROC,                          AUC
> > > > Random head         76.9 (65.5, 87.8)         97.8 (93.8, 99.9)
> > > > Random head         77.2 (66.5, 87.6)         97.5 (93.9, 99.9)
> > > > Random head         75.3 (64.0, 86.6)         97.2 (93.5, 99.8)
> > > >
> > > > As shown, the performance of each individual head is similar, although slightly lower than the baseline model which is indicative of specialisation. Based on these results, the difference between the individual heads and their ensembled prediction also seems bigger compared to the deep ensemble and it’s members. These results suggest a bigger specialisation between the heads compared to the ensemble members.
> > > >
> > > > 2. The reviewer furthermore suggests to include an analysis on model disagreement, following work by Lakshminarayanan. We agree that this metric is valuable when assessing the diversity of the predictions and the performance of each model. Therefore, we propose to include these results in a camera-ready version of the paper.
> > > >
> > > > Here, we use the same definition of model disagreement as proposed by  Lakshminarayanan: the sum over all m (#heads or #ensemble members) of the Jensen-Shannon divergence (JSD) values of p_m with p_model. Here, p_m is the individual prediction and p_model is the ensembled prediction. We multiply this value by 1 over M to compensate for the varying amount of heads or ensemble members.
> > > >
> > > > Similar to our previous analysis, we can try to discriminate in-distribution whole-slide-images (WSIs) from OOD WSIs, but based on model disagreement instead of entropy. Here, we get the following results:
> > > >
> > > > Out-of-distribution detection task (lymphoma detection).
> > > > Model,                  FPR at 95% TPR             AUROC
> > > > Ensemble-5,         42.5 (24.0, 62.0)             85.1 (77.5, 91.9)
> > > > M-heads-5,           38.5 (23.1, 63.0),            87.6 (81.3, 93.2)
> > > >
> > > > As shown, the multi-head model seems to be slightly better at discriminating between	in-distribution and out-of-distribution WSIs based on the JSD values. However, these results are not immediately clear when looking at the corresponding JSD values per model.
> > > >
> > > > Mean (and standard deviation) of JSD values per model.
> > > > Model,                 in-distribution WSIs            out-of-distribution WSIs
> > > > Ensemble-5,        0.7e-3 (+/- 1.0e-3)                3.9e-3 (+/- 3.6e-3)
> > > > M-heads-5,          1.0e-3 (+/- 0.9e-3),               3.7e-3 (+/- 2.6e-3)
> > > >
> > > > Accurately comparing the exact JSD values between the ensemble method and the multi-head approach, however, is difficult due to the differences in the training data used to train each model. For a more fair comparison, a deep ensemble which is trained without bagging should be included in future work. Based on these results, we will update the future work section of the discussion. We would also like to argue that future work should further analyse the diversity within each model. For instance, one could evaluate diversity throughout the (final layers) of these networks.
> > > >
> > > > Due to the limited time and computational resources available, we do not have the results for the other models used in this work currently. However, we will include the entire list in the camera-ready version of this work.

---

> > > > > ### Author Response · Authors · 2020-04-04
> > > > > **Included additional analysis on the behaviour of M-heads (2/2)**
> > > > >
> > > > > 3. Next, the author mentions that the paper could be improved by adding results of a multi-head model with 10 heads, trained without the meta-loss function (similar to the approach we took for the model with five heads).
> > > > >
> > > > > Here we would like to emphasize the efficiency of the multi-head approach. Training and evaluating this additional model was possible, even under the time and resource constraints for this discussion phase. We will include these additional results in the camera-ready version of the paper.
> > > > >
> > > > > Target task performance (breast cancer metastasis detection):
> > > > > Model,                      FROC,                          AUC
> > > > > M-heads-10-min,    75.4 (64.0, 86.6)        97.2 (93.5, 99.8)
> > > > >
> > > > > Out-of-distribution detection task (lymphoma detection).
> > > > > Model,                          FPR at 95% TPR,                          AUROC
> > > > > M-heads-10-min,        41.8 (25.0, 59.1)                           88.4 (82.0, 93.8)
> > > > >
> > > > > In line with our previous results, we see a similar drop in performance for the task of OOD detection when not using the meta-loss.
> > > > >
> > > > > The reviewer also questions the fact that the performance does not show diminishing returns when increasing the number of heads in our work. However, we do expect diminished returns for the multi-head model as well, when increasing the number of heads even further. The fact that this problem arises earlier in deep ensembles could be due to the difference in the number of parameters. Deep ensembles have significantly more parameters, thus issues of redundancy will show up earlier.
> > > > >
> > > > > Next, the reviewer mentions the possibility of training a deep ensemble using the meta-loss function when applying forms of gradient accumulation across batches and replacing batchnorm with other normalisation methods. We concede to this point of the reviewer and agree that it is possible to do so when altering the model. We will update discussion accordingly by stating that it is possible to train a deep ensemble using the meta-loss function although it would require significant changes to the model.
> > > > >
> > > > > Finally, the reviewer includes two detailed comments, about a reference and the use of CDF plots instead of PDF plots. We thank the reviewer for these details and we will include those in the camera-ready version of the paper as well.
> > > > >
> > > > > We thank the reviewer again for his or her time and we are happy that our discussion led to the additional results on model disagreement in the paper. We hope to have convinced the reviewer with these additional results.

---

> ### Comment · AnonReviewer1 · 2020-04-04
> **Reviewer's Final thoughts**
>
> I like to thank the authors for their extensive clarifications and additional results. The additional insights have alleviated my concerns about the experimental results and offered additional insights into the behaviour of regular ensembles and M-Heads. Given the inclusion of those findings I would like to see this paper accepted at MIDL 2020 and change my rating to "3: Weak Accept". However, I seem unable to edit my previous review to reflect this change.

---

### Official Review · AnonReviewer2 · 2020-03-11
**Application of multiple hypothesis prediction (MHP) for uncertainty quantification in DNNs for medical diagnosis.**

**Rating:** 3
**Confidence:** 4
**Recommendation:** Poster

**Summary:**

The paper deals with an important topic: uncertainty quantification in DNNs for medical diagnosis, in particular, digital pathology. It is well motivated and written. However, it is only an application of multiple hypothesis prediction (MHP) models to classification problems in pathology. In this regard, the study is missing out on novelty. But, I liked the approach and how a multi-head network can compete with other more demanding methods like Deep Ensembles.

**Strengths:**

- Decent problem specification and
- A competitive model with less computational demands
- Actually multi-head model performs better than others.
- Comparison with other methods, good evaluation and promising results

**Weaknesses:**

Despite its simplicity and other benefits like improved performance with less computation compared to Deep Ensembles, head diversification introduces an additional hyperparameter. To avoid the mode collapse at the head level, dropout is also added, which adds more hyperparameter(s): dropout rate. Even though I could live with a few or couple of additional hyperparameters, especially considering the aforementioned benefits, it might be some sort of nuisance to deal with additional hypers in other applications. But not too bad.

**Detailed Comments:**

Fig 1:
In summation, are we missing 1/N for the mean prediction ?

Training details:
"we use a fully convolutional Densenet architecture" FCN made me think that you were dealing with segmentation for a second. In the end, patch-based decision are used to obtain heatmaps, which is similar to segmentation but the word 'fully' is misleading.

Results:
FNR is supposed to be FPR, I guess. This is a mistake all over the place.

**Justification Of Rating:**

A decent application of multiple hypothesis prediction (MHP) models. Considering the computational gains on top of performance, it is a promising work. However, there some issues pointed at above.  If they are clarified, then I would like to see this work accepted.

**Paper Type:**

validation/application paper

**Questions To Address In The Rebuttal:**

Section 4.2:
"We observe that the multi-head model mostly displays aleatoric uncertainty for the in-distribution sample through high uncertainty at class boundaries (Der Kiureghian and Ditlevsen, 2009)."
How did we end up with aleatoric uncertainty? There was no mention of different uncertainty types. How could you separate uncertainty types and measure them? These are not clear and hard to believe based on the current form of the paper.
Discussion, page 8:
"We  also  observed  better  calibrated  predictions  for  both  the  multi-head  and  deep  ensembles  approach."
Could you be more specific with some measures? ECE or variants, or reliability diagrams could help. If not, maybe, you could point at Fig.5 and explain more.

"We hypothesize that discriminating between epistemicand aleatoric uncertainty could help to increase the information provided by the uncertaintysignal.  This could improve down-the-line tasks such as out-of-distribution detection."
This goes inline with my earlier comment. How to detect/separate different uncertainty types? Can you tell the different from the mean or weighted head prediction? Or would it be to naive to assume so? From the previous comment, it reads like you have done it already but we cannot see it.

**Special Issue:**

no

---

> ### Author Response · Authors · 2020-03-27
> **Comments for AnonReviewer 2. Improved the submission accordingly.**
>
> We thank the reviewer and we are happy that he or she points out strengths in our problem specification, the evaluation and the corresponding results. However, some concerns are raised which we are happy to respond to. To address some of these comments we have updated our submission accordingly.
>
> First, the reviewer correctly identifies the possible nuisance of additional hyperparameters in the multi-head approach. Although this requires additional efforts in optimisation we would like to point out that this is true for all related uncertainty estimation approaches as well. For example, MC-dropout requires the use of dropout layers. However, it is non-trivial to determine the number of dropout layers, their location within the network and the associated dropout probability. The same is true in Deep Ensembles and the number of members used.
>
> The reviewer furthermore raises questions about the different types of uncertainty that are considered in our discussion section. Reflecting on this review we have decided to remove this part from our discussion section with no impact on the rest of the discussion. To answer the reviewer's question: in this work we only model epistemic uncertainty. However, we still think that modelling aleatoric uncertainty additionally could be beneficial for OOD-detection. Therefore we have included this in our discussion section to motivate future work.
>
> To address the reviewers comment on the point of model calibration we have revised this part of the discussion section into: “Figure 5 (b) shows that ensembles and the multi-head approach are less confident in general, compared to the other methods. Furthermore, we see in Figure 5 (a) that the multi-head approach and the ensembles are more accurate at higher confidence values. In other words, both multi-heads and deep ensembles are less frequently wrong with high confidence. This suggests a better calibration by the deep ensembles, in line with previous work, as well as by the multi-head approach.”. Although we are happy to address any further questions during the discussion period of MIDL as well.
>
> Some of the reviewer's detailed comments are also reflected in the updated submission: The missing 1/M factor is included in Fig 1. Also, any incorrect mentions of FNR are changed to FPR.
>
> Lastly, the reviewer mentioned his or her confusion about the term fully convolutional neural networks. Indeed, we do not directly train a segmentation model as is the case in a U-net for instance. However, we apply a patch-based classification approach, as is common in digital pathology, to produce heatmaps at a whole-slide image level. Here we borrow terminology from a landmark paper by J. Long (2015) on conversion of classification architectures with fully-connected layers to architectures with only convolutional layers..
>
> We thank the reviewer again for his or her comments and we are happy that they led to improvements to our submission. We are open to answer any remaining questions in the discussion period of MIDL.

---

> > ### Comment · AnonReviewer2 · 2020-03-29
> > **Maybe a little more push towards improvements**
> >
> > Thanks for the clarifications and revision. I am happy to see that my comment on the quantification of different uncertainty types has been well received. However, I need to explicitly say that I am actually fine with the contemplated future direction in Discussion:
> > "Furthermore, based on the findings presented in Figure 4, we hypothesize that discriminating between epistemic and aleatoric uncertainty could help to increase the information provided by the uncertainty signal. This could improve down-the-line tasks such as out-of-distribution detection."
> > Of course, understanding the source/cause of uncertainty would tell us useful things and we can take measures based on those. To me, the bigger problem was in Section 4.2:
> > "We observe that the multi-head model mostly displays aleatoric uncertainty for the in-distribution sample through high uncertainty at class boundaries (Der Kiureghian and Ditlevsen, 2009)."
> > If I use my imagination, I can come with a story for this. The model learns from the in-dist examples and reduces its epistemic uncertainty; however, the data-centric aleatoric uncertainty remains. Maybe, the reference also describes such a scenario. However, the reader is not supposed to read each and every reference. You could/should be more explicit here and provide a better sketch. Or, if this is only a speculation, maybe, you should do it Discussion and improve the quality of Results section, with minimal confusion. I think this would improve your paper, if it is done well. Please, take one more shot at it, instead of just removing the relevant parts.
> >
> > At the same time, the uncertainty is measured from the mean prediction from multiple heads. In this regard, I need emphasise that this will be inevitably subject to an overlap between the epistemic and aleatoric uncertainties, especially when there is no mechanism to separate them. In your current approach, there is no way to separate them. Maybe, you can say that you measure "predictive uncertainty" (to be politically correct) and in the future you will study the contributing factors, etc...
> >
> > In addition, please add a footnote saying that  you "borrow terminology from a landmark paper by J. Long (2015)" in order to help your future readers avoid possible confusions.
> >
> > I look forward to seeing the revised manuscript and, of course, your response to my comments.

---

> > > ### Author Response · Authors · 2020-04-04
> > > **Addressing the uncertainty types in the future work section of our discussion**
> > >
> > > We thank the reviewer for his or her final thoughts on our work and we are happy that the reviewer thinks it is of interest to the MIDL community.
> > >
> > > The reviewer raises the issue of mentioning the different types of uncertainty in section 4.2. We mistakenly put “discussion section” in our previous comment. We meant to say: “we have decided to remove this part from our result section with no impact on the rest of the results”.
> > >
> > > We will include the idea of disentangling the predictive uncertainty into epistemic and aleatoric uncertainty in our future work section. Here we will cite work by A. Kendall and  Y. Gal, (“What Uncertainties Do We Need in Bayesian Deep Learning for Computer Vision?”) as a way to do so.
> > >
> > > To address any confusion regarding the term fully convolutional neural networks, we will add the footnote as suggested by the reviewer.
> > >
> > > We thank the reviewer again for his or her comments.

---

> ### Comment · AnonReviewer2 · 2020-04-03
> **Reviewer's Final Thoughts**
>
> I think the paper demonstrates a decent application/validation study of the multi-head (hypotheses) predictor networks. It will be interesting for the MIDL community and attendees.
>
> I keep my initial rating.

---

### Official Review · AnonReviewer3 · 2020-03-19
**The novelty is limited. A comprehensive analyzing on computational cost is needed.**

**Rating:** 2
**Confidence:** 3

**Summary:**

This paper presents a method for out-of-distribution sample detection. They use several convolutional neural networks (heads) to improve the performance of such a task.   Using a set of models or ensemble learning is not a new idea for improving the detection rate.  Furthermore, it seems the proposed method has achieved acceptable performance in the expense of increasing the computational cost.


**Strengths:**

The paper addresses a challenging problem.  They have exploited several models for detecting the out-of-distribution samples.
Experimental results show the proposed method is able to detect the out-of-distribution samples.


**Weaknesses:**


-The novelty is very limited. Using several models (here heads), instead of one, is not a new idea.

-The experimental results are incomplete.   A comprehensive discussion and also comparison to the state-of-the-art method are necessary.

- Multi-head increase computational cost. An analyzing of this term and comparing it with the previous method is needed.

-It is not clear, why such a method works better than one-class classification. It seems the generality of such a method on new samples (unseen samples) be better than this method.

**Justification Of Rating:**

The novelty of paper is not enough.
Experimental results should be improved
The complexity of the method should be analyzed.
The paper ignored mentioning all important previous methods for out-of-distribution detection.


**Paper Type:**

both

**Special Issue:**

no

---

> ### Author Response · Authors · 2020-03-27
> **Comments for AnonReviewer3. Addressed computational costs per model.**
>
> We thank the reviewer for his or her time.  We will respond to the points that are raised in the review, below. We would like the reviewer to clarify some comments as well.
>
> In response to the first comment, it does seem that the reviewer has misinterpreted the model we use, as the reviewer mentions that several models were used instead of one. However, we are not using a standard ensemble of CNNs but instead use a single convolutional neural network with multiple outputs (heads).
>
> With respect to the question on computational complexity, we calculated this for both the baseline model (single model), our multi-head approach (single models, multiple outputs) and the standard ensemble approach (multiple models).
>
> We will only consider convolutional operations since they contribute the most to the computational costs. For each layer, we calculated the number of operations as: kernel-width * kernel-height * input-width * input-height * input-channels * output-channels. We then multiply these operations times two to include both multiplication and summation operations. The sum over all layers approximates the total number of operations per model (#operations). Only considering the operations during the forward pass through the network for simplicity, we end up with the following numbers per input sample:
>
> Single forward pass:
> Model,              #operations
> Vanilla,              2.658 * 1e6
> MC-dropout,     2.658 * 1e6
> Ensemble (5),   5 * 2.658 * 1e6
> Ensemble (10), 10 * 2.658 * 1e6
> M-heads (5),     2.676 * 1e6
> M-heads (10),   2.699 * 1e6
>
> Forward pass including uncertainty estimation:
> Model,               #operations
> Vanilla,              2.658 * 1e6
> MC-dropout,     32 * 2.658 * 1e6
> Ensemble (5),   5 * 2.658 * 1e6
> Ensemble (10), 10 * 2.658 * 1e6
> M-heads (5),     2.676 * 1e6
> M-heads (10),   2.699 * 1e6
>
>
> As shown, the multi-head model only marginally increases the computational costs of a vanilla CNN compared to the other uncertainty estimation methods. It is worth noting that this difference in computational complexity is especially meaningful in the field of digital pathology, where images are expressed in terms of gigapixels.
>
> The reviewer suggests that the result section is incomplete, lacking a comparison with the state-of-the-art. We would like to ask the reviewer to clarify this statement. In this work, we do include a comparison with several state-of-the-art approaches: e.g., MC dropout and deep ensembles. For a recent comparison of related approaches, we refer the reviewer to a paper by Ovadia et al. (2019). In that work, presented at NeuRIPS, the authors show superior results of deep ensembles in an extensive comparison with other approaches on multiple tasks. As such, due to the inclusion of deep ensembles as a comparison, we have included what is currently considered state-of-the-art.
>
> Similarly, the reviewer mentions one-class classification approaches. We are surprised by this comment. Our work focuses on methods capable of predicting a certain target task: p(y|x), which is motivated in both the introduction as well as the related work section. The method suggested by the reviewer falls in an entirely different problem-setting (where you are interested in just p(x)) with a different set of approaches.
>
> Finally, we would like to address the reviewer's comment on the novelty of this work. Following the author instructions provided by the MIDL conference, full papers should contain well-validated applications or methodological contributions. While the underlying model in this work is not new, we aim to get a more rigorous understanding of existing methods of uncertainty estimation in medical imaging, a topic we deem highly valuable by all reviewers. Furthermore, comparative analysis of the multi-head approach with related work is limited and especially lacking in medical imaging. Additionally, our work uniquely adds to the existing body of work on uncertainty estimation by evaluating popular approaches on a clinically relevant and large-scale medical imaging dataset. The results that follow from this work demonstrate superior results by the multi-head approach in terms of both target task and OOD-detection performance.

---

### Meta-Review · Area_Chair1 · 2020-04-05
**MetaReview of Paper140 by AreaChair1**

**Rating:** 3
**Recommendation For Accepted Papers:** Poster

**Metareview:**

Thanks for the reviewers taking time to comment and discuss. The authors' extensive rebuttal successfully convinced the two reviewers (with high-confidence evaluation) in the discussion phase. I also think that the topic of out-of-distribution detection will be interesting for the MIDL community. So an acceptance is suggested. Authors should revise the final version by addressing the issues as discussed with the reviewers.

**Paper Type:**

methodological development

**Special Issue:**

no

---

### Decision · Program_Chairs · 2020-04-11

Accept